# Regulation of Eag1 gating by its intracellular domains

Jonathan R Whicher, Roderick MacKinnon*

Laboratory of Molecular Neurobiology and Biophysics, The Rockefeller University, Howard Hughes Medical Institute, New York, United States

**Abstract** Voltage-gated potassium channels ($K_v$s) are gated by transmembrane voltage sensors (VS) that move in response to changes in membrane voltage. $K_v10.1$ or Eag1 also has three intracellular domains: PAS, C-linker, and CNBHD. We demonstrate that the Eag1 intracellular domains are not required for voltage-dependent gating but likely interact with the VS to modulate gating. We identified specific interactions between the PAS, CNBHD, and VS that modulate voltage-dependent gating and provide evidence that VS movement destabilizes these interactions to promote channel opening. Additionally, mutation of these interactions renders Eag1 insensitive to calmodulin inhibition. The structure of the calmodulin insensitive mutant in a pre-open conformation suggests that channel opening may occur through a rotation of the intracellular domains and calmodulin may prevent this rotation by stabilizing interactions between the VS and intracellular domains. Intracellular domains likely play a similar modulatory role in voltage-dependent gating of the related $K_v11-12$ channels.
DOI: https://doi.org/10.7554/eLife.49188.001

*For correspondence:
mackinn@mail.rockefeller.edu

**Competing interests:** The authors declare that no competing interests exist.

## Introduction

Voltage-gated potassium channels ($K_v$s) conduct potassium ions in response to changes in membrane voltage. All $K_v$s are tetramers and consist of 6 transmembrane segments (S1-S6) (*Jiang et al., 2003*; *Long et al., 2005a*; *Long et al., 2007*; *Sun and MacKinnon, 2017*; *Wang and MacKinnon, 2017*; *Whicher and MacKinnon, 2016*). S1-S4 form the voltage sensor (VS) and S5-S6 form the potassium pore. The S4 helix of the VS is positively charged and was proposed to move within the membrane in response to changes in membrane voltage (*Aggarwal and MacKinnon, 1996*; *Seoh et al., 1996*). Upon membrane hyperpolarization the S4 was predicted to move 'down' towards in the intracellular side of the membrane to close the potassium pore and upon membrane depolarization the S4 was predicted to move 'up' to the extracellular side of the membrane to open the potassium pore. In $K_v$s 1–9 movement of S4 is coupled to the potassium pore by a ~ 15 residue, helical S4-S5 linker, which forms a domain-swapped linkage between S4 and S5 that positions it directly above the pore lining S6 helix (*Long et al., 2005a*; *Long et al., 2007*; *Sun and MacKinnon, 2017*). In this position, the S4-S5 linker was proposed to function as a mechanical lever to couple movement of the S4 to the S6 helices to open and close the pore (*Long et al., 2005b*). However, in $K_v$s 10–12 the S4-S5 linker is only six residues, which is not long enough to form the domain-swapped linkage observed in $K_v$s 1–9. Indeed, recent structures of Eag1 ($K_v10.1$) and hErg ($K_v11.1$) revealed that the S4-S5 linker forms a non-domain swapped linkage between S4 and S5 and is unlikely to be domain swapped in any conformation (*Figure 1a,b,c*) (*Wang and MacKinnon, 2017*; *Whicher and MacKinnon, 2016*). Due to the non-domain swapped transmembrane architecture, the S4-S5 linker is not in the same position and does not appear to form a similar mechanical lever, suggesting an alternative mechanism of voltage-dependent gating in $K_v$s 10–12 (*Lörinczi et al., 2015*; *Tomczak et al., 2017*).

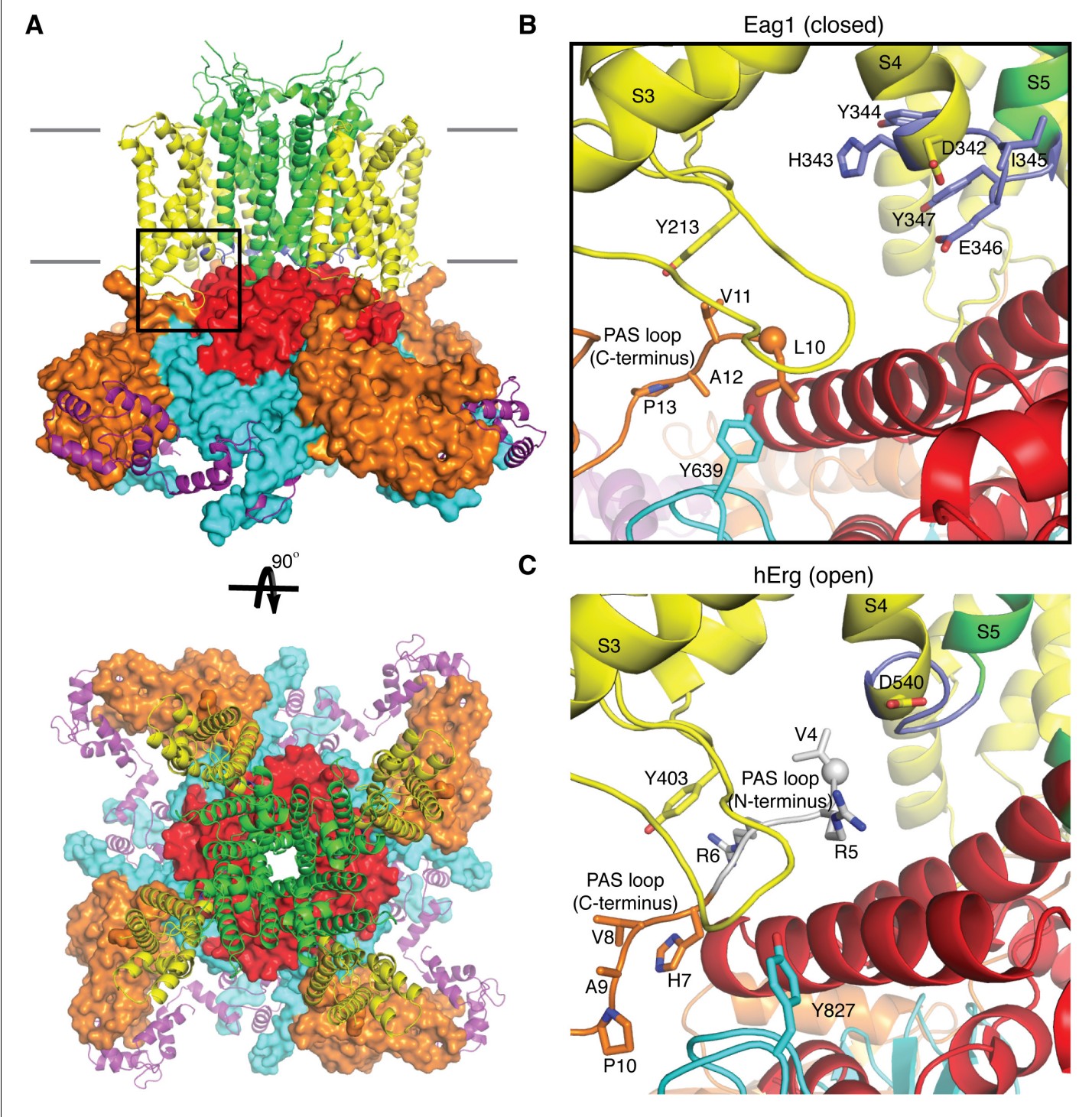

**Figure 1.** Comparison of the PAS loop in Eag1 and hErg. (A) Structure of Eag1 (PDB: 5K7L) in the closed conformation (PAS is orange, VS is yellow, S4-S5 linker is blue, pore is green, C-linker is red, CNBHD is cyan, CaM is purple, and membrane is indicated by gray bars) with a black box indicating the view for panel (B). (B) View of interaction between PAS loop, VS and CNBHD in the closed conformation of Eag1 with the same coloring as in (A). The N-terminus of the protein is shown as a sphere. The PAS loop N-terminus is not observed in this structure. (C) View of the interaction between PAS loop, VS and CNBHD in the open conformation of hErg (PDB: 5VA2) with the same coloring and orientation as in (B). The N-terminus is shown as a sphere and the PAS loop N-terminus is shown in gray.

DOI: https://doi.org/10.7554/eLife.49188.002

$K_v$s 10–12 have three intracellular domains: an N-terminal Per-ARNT-Sim domain (PAS), a C-terminal C-linker domain, and a C-terminal cyclic nucleotide binding homology domain (CNBHD). Interactions between the VS and intracellular domain observed in the structure of Eag1 suggests that the intracellular domains may function in voltage-dependent gating (*Whicher and MacKinnon, 2016*) (*Figure 1a*). For example, the PAS domain, which is positioned directly below the VS, has a 15-residue N-terminal loop (PAS loop) that is directed, through interactions with the VS and CNBHD, towards the S4-S5 linker (*Figure 1b*). The PAS loop has been implicated in voltage-dependent gating and the large Cole-Moore effect observed in Eag1, in which more hyperpolarized (negative) resting membrane potentials result in slower rates of activation (*Cole and Moore, 1960*; *Ju and Wray, 2006*; *Ludwig et al., 1994*; *Terlau et al., 1997*). In addition, the C-linker forms an intracellular ring directly below the S6 helices, which are positioned to couple movements of the intracellular domains to the pore (*Figure 1a*). The C-linker ring is near the S4-S5 linker and the S4, which adopts a depolarized or up conformation in the Eag1 structure (*Whicher and MacKinnon, 2016*). Since the S4-S5 linker of Eag1 appeared unlikely to function as a mechanical lever, we proposed a voltage-dependent gating mechanism in which the S4 helix moves towards the intracellular side of the membrane during hyperpolarization to interact with and rotate the C-linker and S6 helices to close the potassium pore (*Whicher and MacKinnon, 2016*). This proposal was based on structural data alone and awaits further examination by functional analysis.

In addition to voltage-dependent gating, Eag1 is also gated by the calcium sensor calmodulin (CaM) (*Schönherr et al., 2000*; *Ziechner et al., 2006*). CaM binds to Eag1 only in the presence of calcium and holds the pore closed even during membrane depolarization. Each PAS and CNBHD domain has a CaM binding site and thus there are eight binding sites per tetramer. In the Eag1 structure, each CaM molecule occupies two binding sites, one on the PAS and one on the CNBHD, clamping the two domains together (*Figure 1a*) (*Whicher and MacKinnon, 2016*). This binding orientation was shown to be essential for channel inhibition but it is unclear how CaM binding prevents opening of the pore. Here we investigate the role of the intracellular domains in voltage-dependent gating and CaM inhibition. We identify interactions between the VS, PAS loop, and CNBHD that modulate voltage-dependent gating and are essential for CaM inhibition. We provide evidence that VS movement during depolarization may destabilize this interface between the PAS loop and the CNBHD to promote channel opening and find that CaM seems to function by stabilizing this interface to inhibit the channel. Finally, we determine a new structure of an Eag1 channel mutant that is insensitive to CaM inhibition. The structure revealed a pre-open conformation of Eag1 (i.e. a conformation that plausibly lies on the conformational pathway leading to opening) and suggests that channel opening may occur through a rotation of the intracellular domains.

## Results

### Role of the Eag1 intracellular domains in voltage-dependent gating

Based on the previous structure of Eag1, we proposed that membrane hyperpolarization causes S4 to interact with and rotate the C-linker to close the pore. To test this hypothesis, we characterized an Eag1 channel in which the PAS, C-linker, and CNBHD were deleted (Eag1TM). The C-terminal assembly domain (887-962) was included in Eag1TM as this domain is needed for tetramer assembly (*Ludwig et al., 1997*). Eag1TM forms functional channels that are voltage dependent (*Figure 2a,b*), indicating that the intracellular domains are not essential for voltage-dependent gating and that an interaction between the S4 and C-linker is not required to close the potassium pore. However, the intracellular domains do modify the gating kinetics, which was demonstrated by both the Eag1TM construct and the Eag1/hErg chimera where the PAS, C-linker and CNBHD from hErg were inserted onto Eag1TM (*Figure 2a,b*). Eag1TM had a right shifted $V_{0.5}$ of 46 mV (zero-slope on the activation curve was not reached up to 100 mV and therefore an accurate value for $V_{0.5}$ could not be measured) compared to the WT Eag1 (19 mV). In addition, the Eag1/hErg chimera has a $V_{0.5}$ (2.5 mV) that is in between that of WT Eag1 and hErg (−22 mV) and exhibits slow rates of channel closure (deactivation), a characteristic of hErg channels. We also recorded the Cole-Moore effect for each construct by holding the cell at increasing holding potentials, from −190 mV to the voltage of channel activation, and stepping to the same depolarized voltage (*Figure 2c*). To compare the Cole-Moore effect from different mutants we plotted holding potential as a function of current at 10 ms

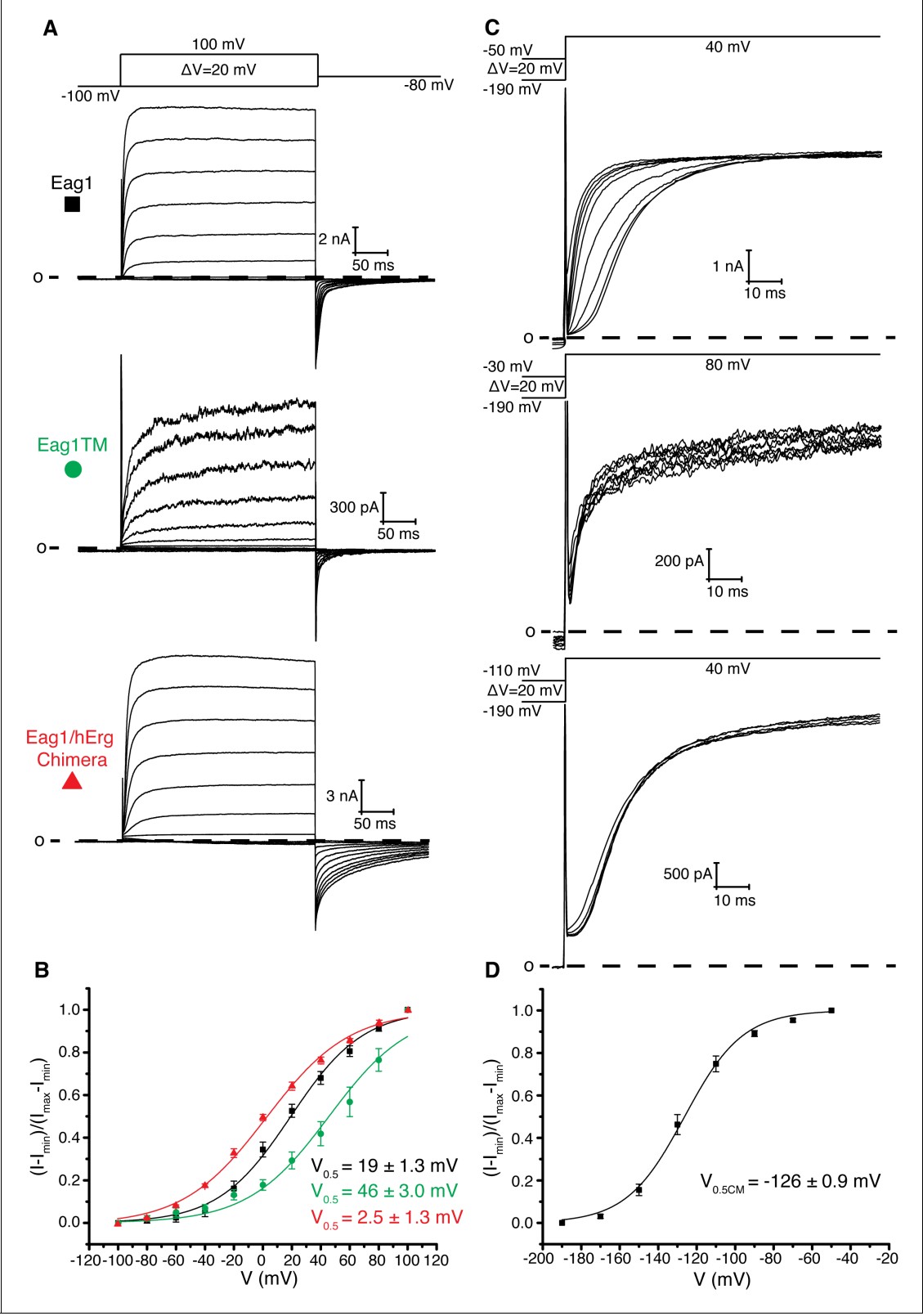

**Figure 2.** Role of intracellular domains in voltage-dependent gating.  (A) Voltage family current trace of WT Eag1, Eag1TM, and the Eag1/hErg chimera with the voltage-pulse protocol shown above. (B) Normalized tail current vs. depolarization voltage plot for WT Eag1 (black square, n = 6), Eag1TM (green circle, n = 4), and the Eag1/hErg chimera (red triangle, n = 4) with $V_{0.5}$ values (mean ± sd). Eag1TM did not reach saturation up to 100 mV. (C) Cole-Moore effect of WT Eag1, Eag1TM, and the Eag1/hErg chimera with the voltage-pulse protocol shown above. (D) Plot of normalized current at 10

*Figure 2 continued on next page*

*Figure 2 continued*

ms following the depolarization step vs holding potential for WT Eag1 (Cole-Moore I-V plot). The Cole-Moore I-V plot was fit with a Boltzmann function to estimate the holding potential that produces half maximal rates of activation ($V_{0.5CM}$ = −126 ± 0.9 mV, mean ± sd, n = 6).

DOI: https://doi.org/10.7554/eLife.49188.003

following the depolarization step. Then we fit the plot with a Boltzmann function (defined in the Materials and methods) to estimate the holding potential that produces half maximal rates of activation ($V_{0.5CM}$) (*Figure 2d*). Neither mutant channel exhibits a Cole-Moore effect (*Figure 2c*), which, along with the slow deactivation of the Eag1/hErg chimera and the shifted $V_{0.5}$ of both channels, demonstrates that the intracellular domains influence voltage-dependent gating kinetics. In addition, this result suggests that the Cole-Moore effect may arise from an interaction between the transmembrane and intracellular domains.

## Interactions between the voltage sensor and intracellular domains

To search for contacts between the intracellular domains and the transmembrane domains that influence gating properties we searched for mutations that modified or resulted in the loss of the Cole-Moore effect in Eag1. Using the structures of Eag1 and hErg as a guide, we first modified by alanine scanning mutagenesis the C-terminus of S4 and the S4-S5 linker (residues 343–348), which are near the PAS loop in both structures (*Figure 1b*, *Figure 3*, and *Figure 3—figure supplement 1a,b*) (*Gianulis et al., 2013*; *Wang et al., 1998*). These results suggest that the S4-S5 linker plays a role in voltage-dependent gating as we observed both negative (H343A, Y344A) and positive (D342A, I345A, E346A) shifted $V_{0.5}$ values. Furthermore, H343A, Y344A, I345A, E346A, and Y347A all exhibit a Cole-Moore effect with both negative (H343A, I345A) and positive (E346A, Y347A) shifts in the $V_{0.5CM}$ (*Figure 3—figure supplement 1c,d*). We note that only mutation of Asp 342 to Ala (D342A) results in complete loss of the Cole-Moore effect, suggesting that Asp 342 may interact with the intracellular domains (*Figure 3*).

Asp 342 is located at the C-terminus of S4 and is highly conserved in $K_v$s 10–12. In the closed conformation of Eag1, Asp 342 does not interact with the intracellular domains (*Figure 1b*) (*Whicher and MacKinnon, 2016*). However, in the open conformation of hErg the homologous Asp is near (~6 Å) two Arg residues in the PAS loop (*Figure 1c*) (*Wang and MacKinnon, 2017*). In all $K_v$s 10–12 the PAS loop has at least one positively charged residue. In Eag1, the corresponding Arg residues are Arg 7 and 8. We mutated the Arg residues to Ala (R7A/R8A) and deleted residues 3–9 (Δ3–9) (*Figure 3*). The $V_{0.5}$ of R7A/R8A and Δ3–9 are right shifted to a similar extent as Eag1TM and the D342A mutant. Therefore, Arg 7 and 8 promote channel opening, like Asp 342, and mutation of these residues has a similar effect on $V_{0.5}$ as loss of the intracellular domains. In addition, the R7A/R8A and Δ3–9 mutations result in a right-shifted $V_{0.5CM}$, indicating that more depolarized holding potentials are required for fast activation of these mutants than WT Eag1. The right shifted $V_{0.5}$ and the modified Cole-Moore effect of R7A/R8A and Δ3–9 suggest that Arg 7 and 8 might form a functional interaction with Asp 342. Furthermore, since mutation of Arg 7 and 8 did not result in complete loss of the Cole-Moore effect, Asp 342 likely interacts with additional residues on the intracellular domains. Taken together, the functional data along with the proximity of the PAS loop and Asp 342 in the open conformation of hErg suggest that an interaction between Asp 342 and the PAS loop may occur in the open conformation to promote channel opening.

## Implications for voltage-dependent gating

How might an interaction between the PAS loop and Asp 342 promote channel opening? One hypothesis is that the interaction between the PAS loop and Asp 342 stabilizes the depolarized state of the voltage sensor. This hypothesis would explain why deletion of the intracellular domains (Eag1TM) and mutation of Asp 342 and Arg 7 and 8 disfavors channel opening as indicated by a right shift in the voltage-dependence of activation. However, further analysis of PAS loop residues that are adjacent to Arg 7 and 8, Leu 10, Val 11, Ala 12, and Pro 13, revealed an additional function of the PAS loop in voltage-dependent gating. Leu 10, Val 11, Ala 12, and Pro 13 interact with Tyr 639, a conserved residue in $K_v$s 10–12 that is located on the CNBHD, and Tyr 213, a Phe, Tyr or Cys in $K_v$s 10–12 that is located on the loop before S1 of the VS (*Figure 1b*) (*Whicher and MacKinnon,*

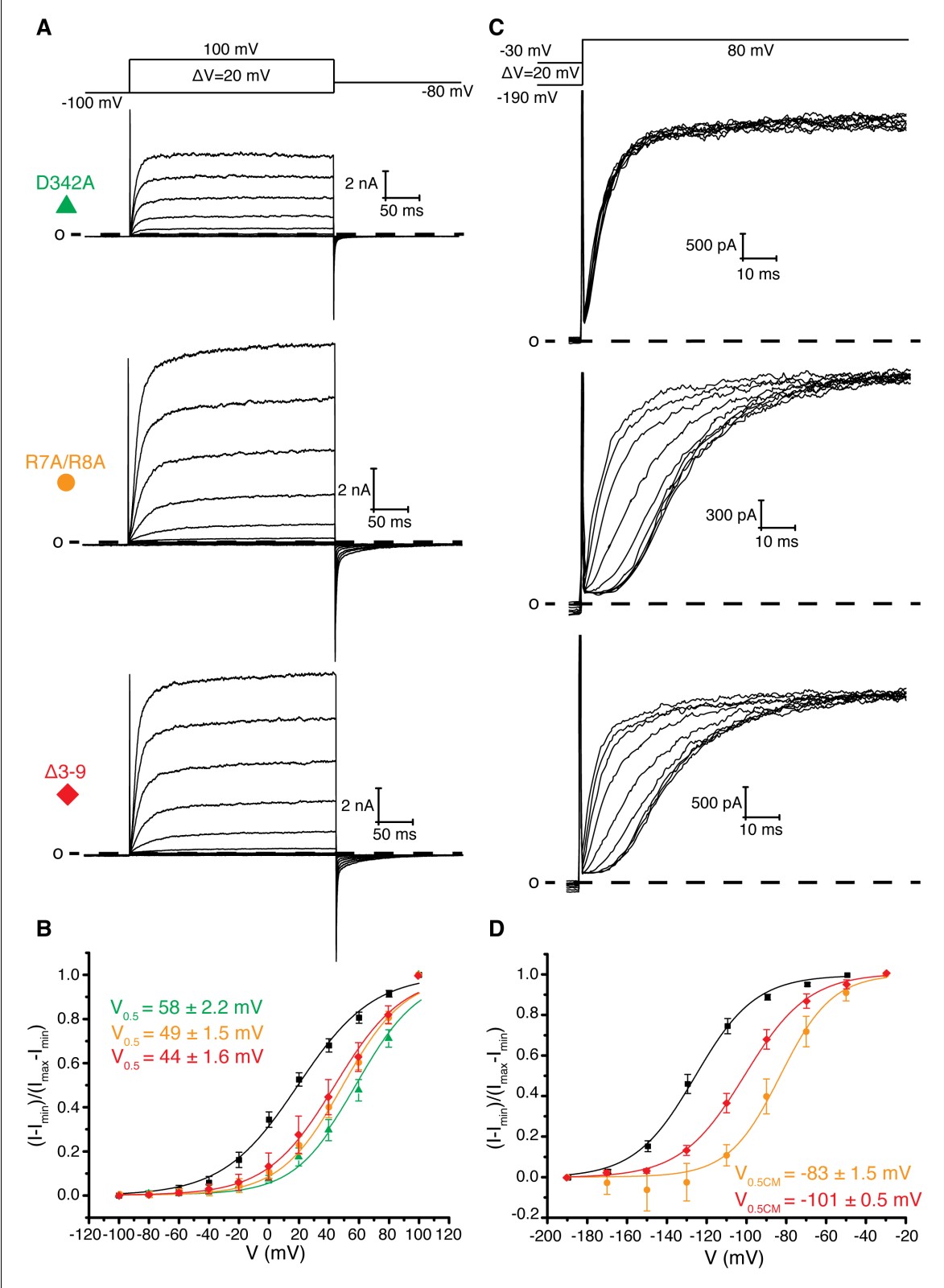

**Figure 3.** Role of Arg 7, Arg 8, and Asp 342 in voltage dependent gating. (**A**) Voltage family current trace for D342A, R7A/R8A, and Δ3–9 with the voltage-pulse protocol shown above. (**B**) Normalized tail current vs. depolarization voltage plot of WT Eag1 (black square, n = 6), D342A (green triangle, n = 5), R7A/R8A (orange circle, n = 5), and Δ3–9 (red diamond, n = 5) with $V_{0.5}$ values (mean ± sd). D342A, R7A/R8A, and Δ3–9 did not

*Figure 3 continued on next page*

*Figure 3 continued*

reach saturation up to 100 mV. (**C**) Cole-Moore effect of D342A, R7A/R8A, and Δ3–9 with the voltage-pulse protocol shown above. (**D**) Cole-Moore I-V plot for WT Eag1 (black square, n = 6), R7A/R8A (orange cirlce, n = 5), and Δ3–9 (red diamond, n = 5) with $V_{0.5CM}$ values (mean ± sd).

DOI: https://doi.org/10.7554/eLife.49188.004

The following figure supplement is available for figure 3:

**Figure supplement 1.** S4-S5 linker mutations.

DOI: https://doi.org/10.7554/eLife.49188.005

*2016*). In this position, residues 10–13 link the VS and the intracellular CNBHD. To study the functional consequences of altering this region of contact, we generated Tyr 213 to Ala (Y213A) and Tyr 639 to Arg (Y639R) mutant channels and PAS loop mutant channels with successive deletions of residues 10–13 (Δ3–10, Δ3–11, Δ3–12, and Δ3–13). These mutants show inactivation and hooked tail currents at more depolarized potentials (40–100 mV), as was previously shown (*Terlau et al., 1997*) (*Figure 4a,b* and *Figure 4—figure supplement 1a,b*). These mutants also produce channels that open at more negative (hyperpolarized) potentials than WT Eag1 (−80 mV) and exhibit slow deactivation, demonstrating that the interaction between residues 10–13, Tyr 213, and Tyr 639 promotes the closed state of Eag1.

Based on these data, the PAS loop can be divided into two functionally distinct segments: the N-terminus (residues 1–9; not observed in the Eag1 structure) and the C-terminus (residues 10–13) (*Figure 1c*). The N-terminus seems to promote channel opening and may interact with Asp 342 of the S4-S5 linker. The C-terminus seems to promote channel closure and interacts with the CNBHD. Furthermore, in the open state structure of hErg the PAS loop C-terminus does not interact with either Tyr 403 (equivalent to Tyr 213 in Eag1) or Tyr 827 (equivalent to Tyr 639 in Eag1) (*Figure 1c*) suggesting that destabilization of this interface might be necessary for channel opening (*Wang and MacKinnon, 2017*). Therefore, we propose that the following structural interactions take place in association with voltage dependent gating. Upon depolarization, Asp 342 interacts with the PAS loop N-terminus to stabilize the open state of the VS and destabilize the interaction between the PAS loop C-terminus, Tyr 213, and Tyr 639 to promote channel opening (*Figure 1c*). When the VS is hyperpolarized, movement of the S4 disrupts the interaction between Asp 342 and the PAS loop N-terminus, allowing the interaction between the PAS loop C-terminus, Tyr 213, and Tyr 639 to form and promote channel closing (*Figure 1b*).

Two experiments support this hypothesis for the role of the intracellular domains in voltage-dependent gating. First, co-expression of two halves of a split Eag1 construct (L341 split), in which the N-terminal half of the channel includes Met 1-Leu 341 and the C-terminal half includes Asp 342-Ser 962, produces a partially constitutively open channel (*Figure 5a*) (*Tomczak et al., 2017*). This functional behavior can be explained by the above hypothesis because in the split Eag1 construct Asp 342 is no longer connected to S4 of the VS and thus is no longer forced to move when the VS moves. Consequently, Asp 342 can maintain its interaction with the PAS loop to promote channel opening. Therefore, if Arg 7 and 8 and Asp 342 functionally interact to promote channel opening then mutation of this interface should produce a split channel that is no longer constitutively open. In agreement with this conclusion, introduction of the D342A and Δ3–9 mutations into the Eag1 split construct produce channels that close at hyperpolarized potentials (*Figure 5b,c*) (*Tomczak et al., 2017*). Second, this mechanism provides an explanation for the Cole-Moore effect. The Cole-Moore effect was proposed to be due to the existence of multiple closed states that the VS must transition through in order to reach an active or depolarized conformation (*Cole and Moore, 1960*). At more negative potentials the VS must transition through more closed states to reach an active conformation, which results in slower activation times. In Eag, at more negative holding potentials the VS might have to transition through more closed states in order for the S4 and Asp 342 to interact with the PAS loop N-terminus, which will result in slower activation times. Therefore, if we remove the interaction between the PAS loop C-terminus, Tyr 213, and Tyr 639, which we propose is destabilized when Asp 342 interacts with the PAS loop N-terminus, then the Cole-Moore effect should be lost. In agreement with this line of reasoning, Δ3–12 and Δ3–13 do not exhibit a Cole-Moore effect while Y213A, Y639R, Δ3–10, and Δ3–11 show a reduced Cole-Moore effect compared to the WT channel (*Figure 4c* and *Figure 4—figure supplement 1c*).

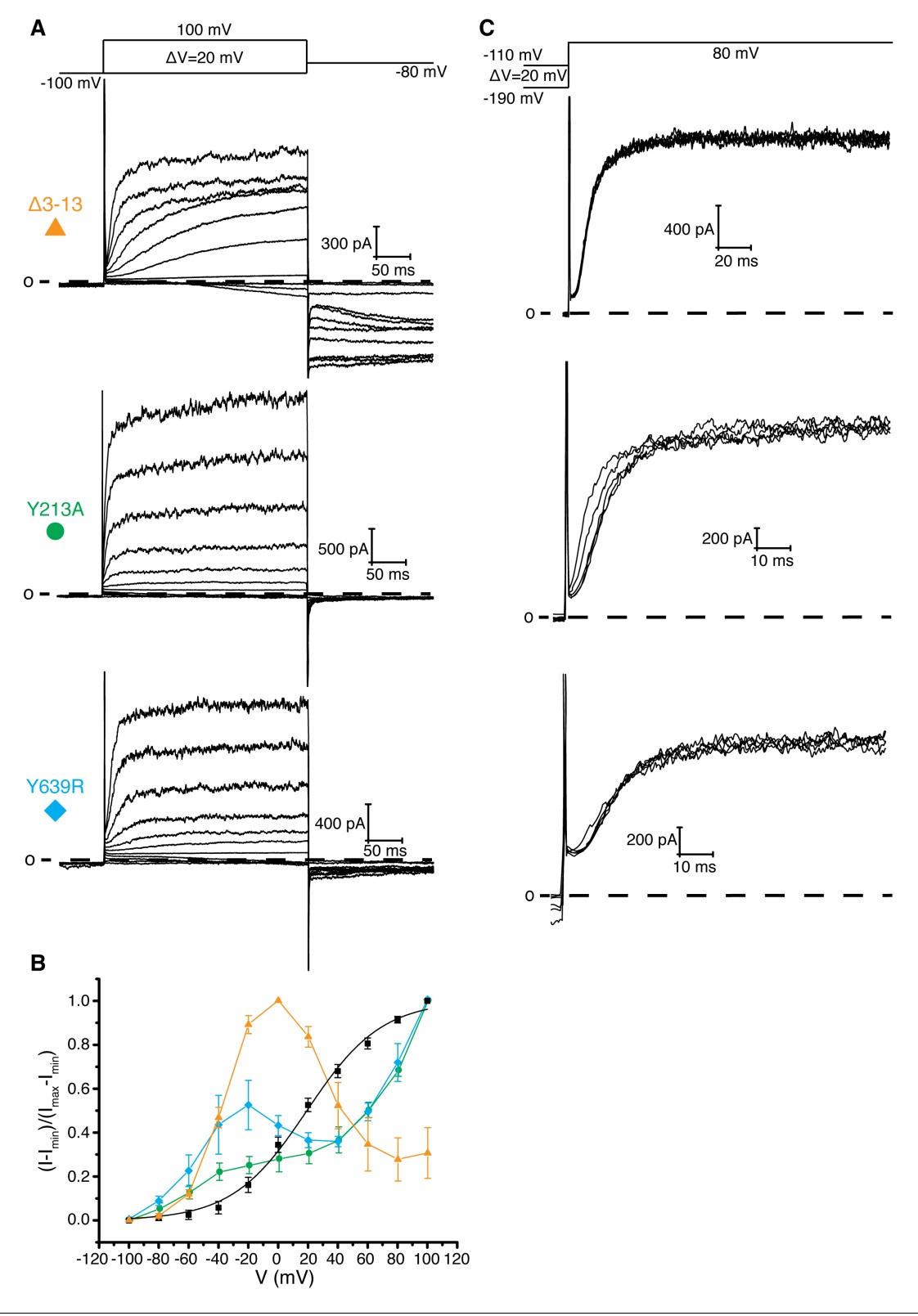

**Figure 4.** Interaction between residues 10–13, Tyr 213, and Tyr 639. (A) Voltage family current trace for the Δ3–13, Y213A, and Y639R with the voltage-pulse protocol shown above. (B) Normalized tail current vs. depolarization voltage plot of WT Eag1 (black square, n = 6), Δ3–13 (orange triangle, n = 5), Y213A (green circle, n = 5), and Y639R (cyan diamond, n = 5) (mean ± sd). (C) Cole-Moore effect of Δ3–13, Y213A, and Y639R with the voltage-pulse protocol shown above.

*Figure 4 continued on next page*

*Figure 4 continued*

DOI: https://doi.org/10.7554/eLife.49188.006

The following figure supplement is available for figure 4:

**Figure supplement 1.** PAS loop deletions.

DOI: https://doi.org/10.7554/eLife.49188.007

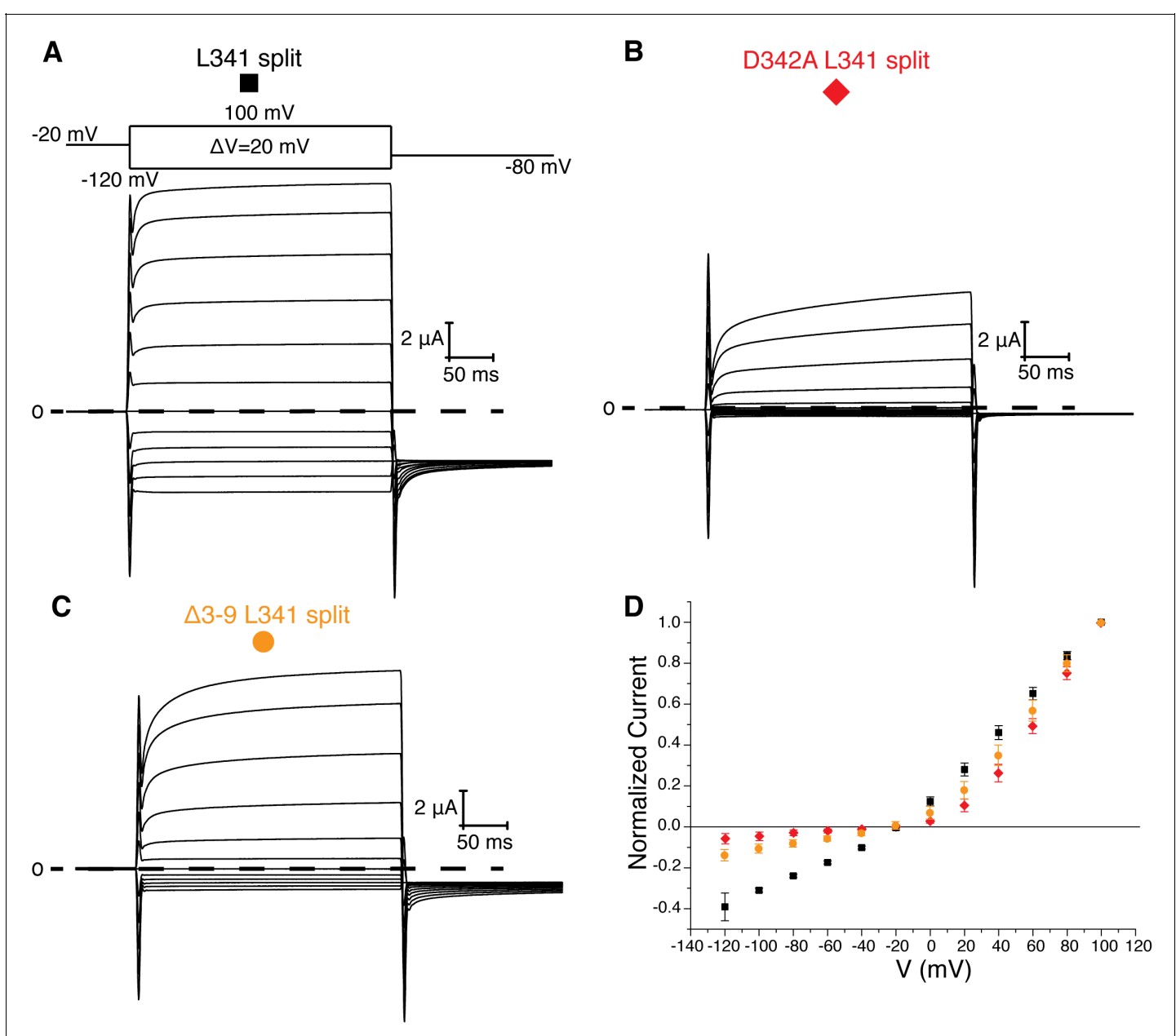

**Figure 5.** L341 split channels. Voltage family current trace for the L341 split (**A**), D342A L341 split (**B**), and Δ3–9 L341 split (**C**) with the voltage-pulse protocol shown above. (**D**) Normalized current vs depolarization voltage for L341 split (black square, n = 11), D342A L341 split (red diamond, n = 7), and Δ3–9 L341 split (orange circle, n = 7) (mean ± sd).

DOI: https://doi.org/10.7554/eLife.49188.008

## Structure of constitutively open Eag1

To better understand how the channel opens and how CaM inhibits opening we sought to determine the structure of an open Eag1 channel. In the presence of $Ca^{2+}$/CaM, Eag1 Δ3–13 remains open at hyperpolarized voltages (*Figure 6a*). Therefore, we determined the Cryo-EM structure of Eag1 Δ3–13 in the presence of calcium and bound to CaM (Eag1 Δ3–13/CaM) (*Figure 6b,c* and *Figure 6—figure supplements 1*, *2*, *3* and *4*). Two different conformations were identified for Eag1 Δ3–13/CaM: conformation 1 at 3.7 Å and conformation 2 at 4.0 Å resolution. In both conformations, the S4 helices adopt a depolarized conformation and intracellular domains are rotated in a counterclockwise direction when viewed from the extracellular side of the membrane, but in conformation two the extent of the rotation is larger (2.4° degrees for conformation 1 and 8.6° degrees for conformation 2) (*Figure 6b,c* and *Figure 6—figure supplement 4*). The rotation observed in these conformations is in a similar direction as the intracellular domains in the open conformation structure of hErg (*Wang and MacKinnon, 2017*) (*Figure 6d*). However, the extent of the rotation of the Eag1 intracellular domains is not as large as the 20° rotation observed in hErg and the S6 helices remain closed (*Figure 6d,e*). As a result, we believe that conformation 1 and conformation 2 represent pre-open conformations of Eag1 on the pathway from closed to fully open.

A hypothesis for why the pore remains closed in the structure of Eag1 Δ3–13/CaM is that, compared to hErg, Eag1 might be more stable in a closed conformation. This hypothesis is consistent with a number of observations on the function. First, insertion of the intracellular domains of hErg onto Eag1 (Eag1/hErg chimera) causes a 20 mV left shift in the $V_{0.5}$ (*Figure 2a,b*). Second, the $V_{0.5}$ of Eag1 is right-shifted by 40 mV compared with hErg and Eag1TM is right shifted by 80 mV when compared to a hErg channel lacking the intracellular domains (*Figure 2a,b*) (*Hausammann and Grütter, 2013*; *Wang and MacKinnon, 2017*). What might cause Eag1 to be more stable in a closed conformation? In Eag1, Phe 475 and Gln 477 are located at the interface of the S6 helices on either side of Gln 476, the intracellular gate. In $K_v$11 and $K_v$12, which have a left shifted $V_{0.5}$ compared to Eag1, these residues are Ile and Arg respectively (*Bauer and Schwarz, 2018*). The Eag1 double mutant F475I/Q477R causes a 50 mV left shift in the $V_{0.5}$ to −20 mV when introduced into the full-length channel and a 16 mV left shift in the $V_{0.5}$ to 30 mV when introduced into Eag1TM (*Figure 7a,b,c*). In addition, when the F475I/Q477R mutation is introduced into the Eag1/hErg chimera the channel remains open at hyperpolarized potentials (*Figure 7d*). Taken together, these data suggest that the intracellular domains and Phe 475 and Gln 477 cause Eag1 to be more stable in a closed conformation. Therefore, we propose that the Eag1 intracellular domains, when viewed from the extracellular side, rotate in a counterclockwise direction to promote the opening of the pore. However, due to the stability of Eag1 in a closed conformation and the conditions under which the Cryo-EM structure was determined we suspect that pore opening is transient and thus not observed in the Eag1 Δ3–13/CaM structure.

## CaM inhibition

Rotation of the intracellular domains observed in the different conformations of Eag1 Δ3–13/CaM occurs with CaM bound to the channel in the same orientation as observed in the WT structure (*Figure 6—figure supplement 4*) (*Whicher and MacKinnon, 2016*). This finding suggests that binding of CaM does not clamp the PAS and CNBHD domains together to prevent rotation of the intracellular domains and channel opening as we previously proposed. Instead, we think that more likely CaM binding to the channel helps to stabilize the hydrophobic interaction between the PAS loop C-terminus, Tyr 213, and Tyr 639. As discussed above, this interaction seems to be important for channel closure. Thus, by stabilizing this interaction CaM can inhibit the channel. In support of this hypothesis, deletion of this interaction by removing residues 3–13 results in a channel that is no longer inhibited by CaM (*Figure 6a*). In addition, a similar effect was observed in Eag1 mutants that lack the PAS cap (residues 1–26) or the entire PAS domain (27-135) (*Lörinczi et al., 2016*).

## Discussion

In summary, the data presented here provide many insights into the mechanism of voltage-dependent gating and calmodulin inhibition in Eag1. First, the intracellular domains are not required for voltage-dependent gating but do modulate voltage-dependent gating kinetics and cause the Cole-

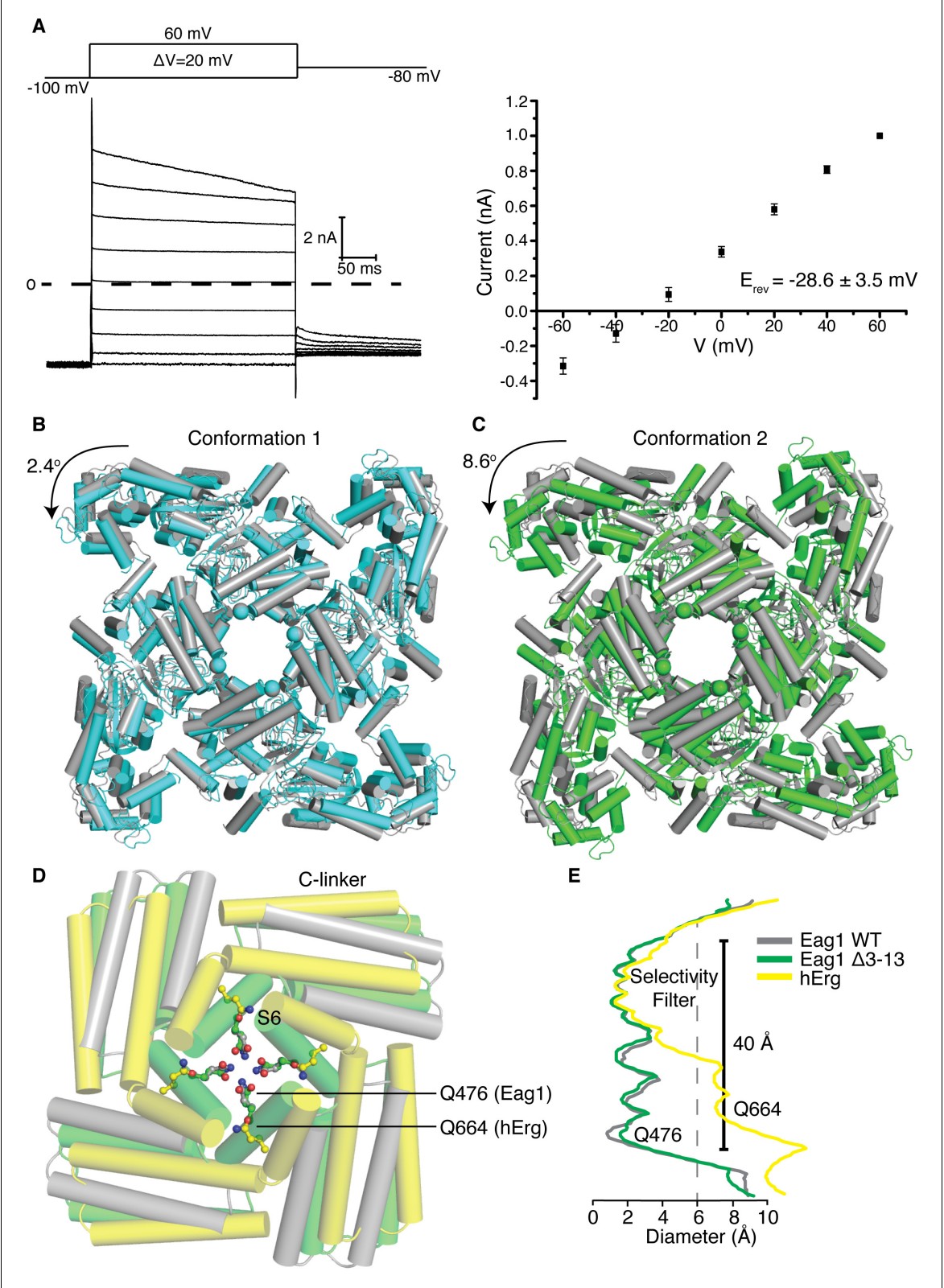

**Figure 6.** Structure of Eag1 Δ3–13/CaM. (**A**) Left, Voltage family current trace for Eag1 Δ3–13 in the presence of 1 mM CaCl$_2$ with the voltage-pulse protocol shown above. Right, normalized current vs depolarization voltage for Eag1 Δ3–13 in the presence of 1 mM CaCl$_2$ (black square, n = 3) with reversal potential (E$_{rev}$) (mean ± sd). (**B**) Structural superposition of Eag1 Δ3–13/CaM conformation 1 (cyan) and Eag1/CaM (PDB-5K7L, (gray) using the selectivity filter. Only the intracellular domains are shown from an extracellular view and the location of the S6 helices are indicated with spheres.
*Figure 6 continued on next page*

*Figure 6 continued*

Degree of rotation is indicated by the arrow. (C) Structural superposition of Eag1 Δ3–13/CaM conformation 2 (green) and Eag1/CaM (gray) using the selectivity filter with the same view as (B). (D) Structural superposition of Eag1 Δ3–13/CaM conformation 2 (green C, red O, blue N), Eag1/CaM (gray C, red O, blue N), and hErg (PDB-5VA2, yellow C, red O, blue N) using the selectivity filter. Location of the intracellular gate Gln (Q476 for Eag1 and Q664 for hErg) are shown as ball and stick. (E) Plot of pore diameter for Eag1 Δ3–13/CaM conformation 2 (green), Eag1 (gray), and hErg (yellow). The location of the selectivity filter and intracellular gate are indicated and the dashed gray line at 6 Å indicates the diameter of hydrated potassium.

DOI: https://doi.org/10.7554/eLife.49188.009

The following figure supplements are available for figure 6:

**Figure supplement 1.** Single-particle cryo-EM structure determination of Eag1 Δ3–13/CaM.

DOI: https://doi.org/10.7554/eLife.49188.010

**Figure supplement 2.** Cryo-EM density map of Eag1 Δ3–13/CaM.

DOI: https://doi.org/10.7554/eLife.49188.011

**Figure supplement 3.** Structure validation of the atomic model of Eag1 Δ3–13/CaM.

DOI: https://doi.org/10.7554/eLife.49188.012

**Figure supplement 4.** Structural superposition of the intracellular domains.

DOI: https://doi.org/10.7554/eLife.49188.013

Moore effect. Second, an interaction between Asp 342 at the intracellular side of the S4 helix and the intracellular domains is essential for the Cole-Moore effect. Through mutagenesis in full length and split channels, we identified Arg 7 and 8 in the PAS loop N-terminus as potential interaction partners for Asp 342. However, mutation of Arg 7 and 8 results in modification but not complete loss of the Cole-Moore effect, suggesting that Asp 342 may interact with other residues on the intracellular domains. Third, interaction between the PAS loop C-terminus, Tyr 213 of the VS, and Tyr 639 of the CNBHD plays an important role in gating of Eag1. Deletion of residues 3–13 from the PAS loop produces a channel that opens at more hyperpolarized potentials than WT Eag1 and has slow deactivation kinetics, suggesting that this interaction promotes channel closure. In addition, this interface is important for the mechanism of CaM inhibition as Eag1 Δ3–13 is constitutively open in the presence of $Ca^{2+}$/CaM. Finally, the structure of Eag1 Δ3–13 bound to CaM in a pre-open conformation demonstrates that channel opening may occur through a rotation of the intracellular domains in a counterclockwise direction when viewed from the extracellular side of the membrane.

Based on the data, we propose the following general mechanism of modulation of voltage-dependent gating by the intracellular domains. In the depolarized conformation of the VS, Asp 342 interacts with the PAS loop N-terminus to stabilize the depolarized state of the VS as well as destabilize the interaction between the PAS loop C-terminus, Tyr 213, and Tyr 639 to promote channel opening (*Figure 1c*). In the hyperpolarized conformation of the VS, movement of the S4 disrupts the interaction between Asp 342 and the PAS loop N-terminus to allow for the interaction between the PAS loop C-terminus, Tyr 213, and Tyr 639 to form and promote channel closing (*Figure 1b*). The structure of Eag1 Δ3–13/CaM in a pre-open conformation suggests that channel opening may occur through a counterclockwise rotation of the intracellular domains and channel closing may occur through a clockwise rotation of the intracellular domains. This mechanism is consistent with previous functional data examining split Eag1 channels that do not have a covalent linkage between the VS and pore and provides an explanation for the Cole-Moore effect (*Lörinczi et al., 2015*; *Tomczak et al., 2017*). At more negative holding potentials the S4 and Asp 342 may have to transition through more closed states in order to interact with the PAS loop N-terminus, which will result in slower activation times. Furthermore, CaM seems to function through this proposed mechanism by stabilizing the interaction between residues 10–13, Tyr 213, and Tyr 639 to prevent pore opening. Perhaps by binding to the PAS domain, CaM may be able to stabilize the PAS loop in a closed conformation.

These interactions also help to understand the gating of related channels. For hErg, which does not exhibit a Cole-Moore effect, the PAS loop Arg residues (Arg 4 and 5) and the S4 Asp (Asp 540) function in slow deactivation (*Morais Cabral et al., 1998*; *Muskett et al., 2011*; *Ng et al., 2011*; *Sanguinetti and Xu, 1999*). How do equivalent residues in Eag1 and hErg function in different voltage-dependent gating outcomes? As discussed above, in Eag1 deletion of the intracellular domains or mutation of Asp 342 causes a positive shift in the $V_{0.5}$, suggesting that the interaction between the PAS loop N-terminus and Asp 342 is important for channel activation and thus functions in the

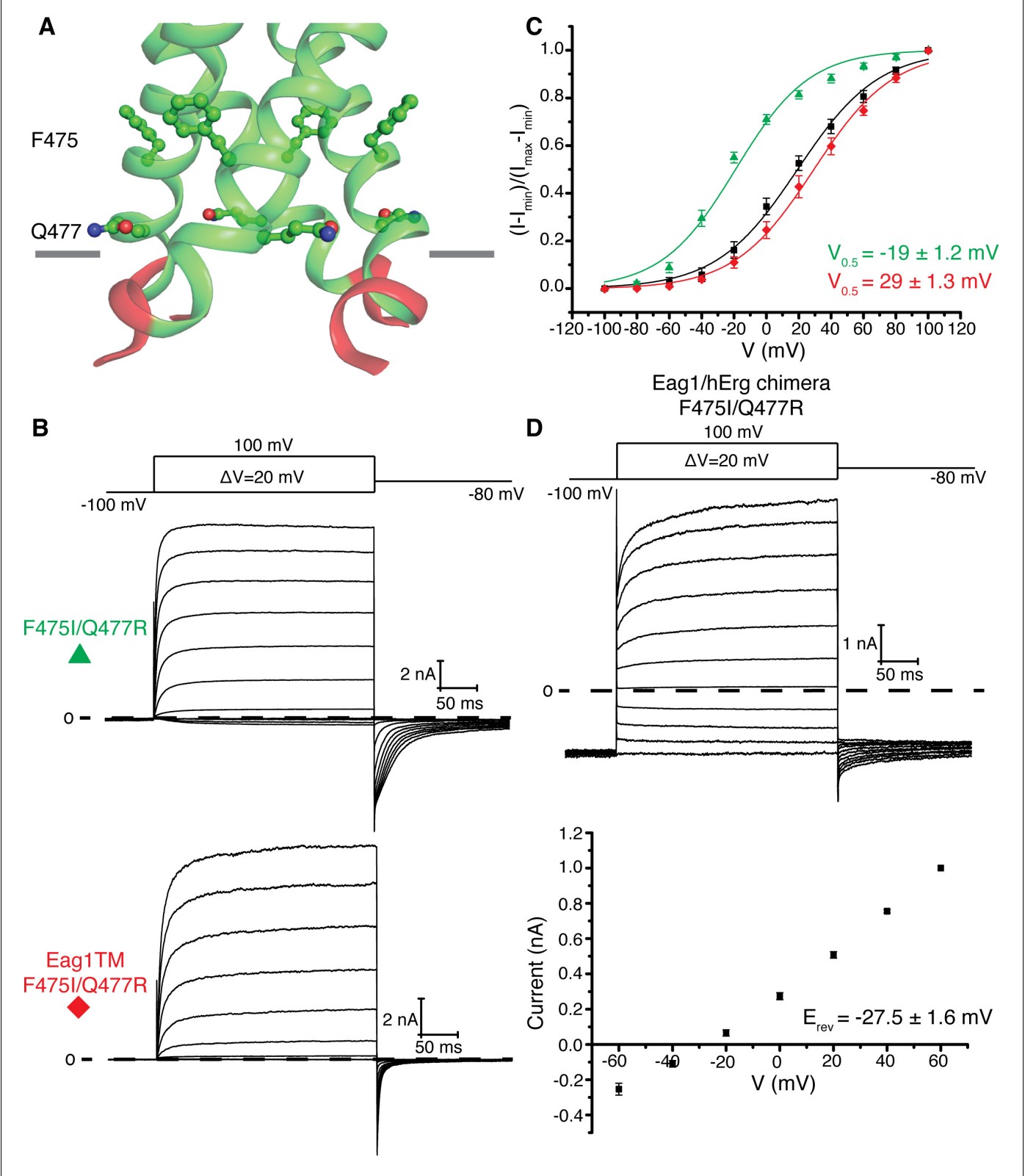

**Figure 7.** Eag1 pore mutants. (**A**) Phe 475 and Gln 477 (shown as green sticks, with red O, and blue N) are at the interface of the S6 helices (green). C-linker is shown in red. (**B**) Voltage family current trace for F475I/Q477R and Eag1TM F475I/Q477R with the voltage-pulse protocol shown above. (**C**) Normalized tail current vs. depolarization voltage plot of WT Eag1 (black square, n = 6), F475I/Q477R (green triangle, n = 7), and Eag1TM F475I/Q477R (red diamond, n = 6) with $V_{0.5}$ values (mean ± sd). (**D**) Top, Voltage family current trace for Eag1/hErg chimera F475I/Q477R with the voltage-pulse

*Figure 7 continued on next page*

*Figure 7 continued*

protocol shown above. Bottom, normalized current vs depolarization voltage for Eag1/hErg chimera F475I/Q477R (black square, n = 5) with reversal potential ($E_{rev}$) (mean ± sd).

DOI: https://doi.org/10.7554/eLife.49188.014

Cole-Moore effect. However, in hErg deletion of the intracellular domains and mutation of Asp 540 does not change the $V_{0.5}$, suggesting that the interaction between the PAS N-terminus and Asp 540 is not important for channel activation (*Hausammann and Grütter, 2013*; *Morais Cabral et al., 1998*; *Sanguinetti and Xu, 1999*). Instead, we propose that in hErg, which we have shown apparently has a more stable open conformation than Eag1, the interaction between Asp 540 and the PAS loop forms after channel activation and prevents pore closing, which results in slow deactivation. Like Eag1, the interaction between Asp 540 and the PAS loop N-terminus in hErg may prevent pore closing by destabilizing the interaction between the PAS loop C-terminus, Tyr 403, and Tyr 827. In support of this idea, splitting the hErg channel, analogous to the Eag1 L341 split, so that Asp 540 is not covalently linked to the voltage sensor, produces a channel that displays slowed deactivation compared to WT hErg (*de la Peña et al., 2018*).

# Materials and methods

## Key resources table

| Reagent type (species) or resource | Designation | Source or reference | Identifiers | Additional information |
|---|---|---|---|---|
| Gene (*Rattus norvegicus*) | K$_v$10.1/Eag1/Kcnh1 | Synthetic | Uniprot: Q63472 | |
| Gene (*Homo sapiens*) | Calmodulin | Synthetic | Uniprot: P0DP24 | |
| Cell line (*Homo sapiens*) | HEK293S GnTI⁻ | ATCC | ATCC: CRL-3022 RRID:CVCL_A785 | |
| Cell line (*Spodopterafrugiperda*) | Sf9 | ATCC | ATCC: CRL-1711 RRID:CVCL_0549 | |
| Cell line (*Cricetulus griseus*) | Chinese Hamster Ovary cells | Sigma | RRID: CVCL_0213 | |
| Recombinant DNA reagent | pEG Bacmam | doi: https://doi.org/10.1038/nprot.2014.173 | | |
| Recombinant DNA reagent | pGEM-T vector | Promega | Catalog number: A1360 | |
| Software, algorithm | pClampfit 10.5 | Molecular Devices | RRID: SCR_011323 | |
| Software, algorithm | MotionCor2 | DOI: 10.1038/nmeth.4193 | RRID: SCR_016499 | http://msg.ucsf.edu/em/software/motioncor2.html |
| Software, algorithm | CTFFIND4 | DOI: 10.1016/j.jsb.2015.08.008 | RRID: SCR_016732 | http://grigorieflab.janelia.org/ctffind4 |
| Software, algorithm | RELION-3 | DOI: 10.1016/j.jsb.2012.09.006 | RRID: SCR_016274 | https://www2.mrc-lmb.cam.ac.uk/relion/index.php?title=Main_Page |
| Ssoftware, algorithm | ResMap | doi: 10.1038/nmeth.2727 | | http://resmap.sourceforge.net |
| Software, algorithm | Coot | doi: 10.1107/S0907444910007493 | RRID: SCR_014222 | https://www2.mrc-lmb.cam.ac.uk/personal/pemsley/coot/ |
| Software, algorithm | Phenix | doi: 10.1107/S0907444909052925 | RRID: SCR_014224 | http://phenix-online.org/ |
| Software, algorithm | Pymol | PyMOL Molecular Graphics System, Schrödinger, LLC | RRID: SCR_000305 | http://www.pymol.org/ |

*Continued on next page*

*Continued*

| Reagent type (species) or resource | Designation | Source or reference | Identifiers | Additional information |
|---|---|---|---|---|
| Software, algorithm | UCSF Chimera | UCSF Resource for Biocomputing, Visualization,and Bioinformatics | RRID: SCR_004097 | http://plato.cgl. ucsf.edu/chimera/ |
| Software, algorithm | HOLE | doi: 10.1016/S0263-7855(97)00009-X | | http://www.holeprogram.org |

## Cloning of Eag1 constructs

Cloning of rat Eag1 into the BacMam (*Goehring et al., 2014*) expression vector with a C-terminal green fluorescent protein (GFP)-His$_6$ tag was described previously (*Whicher and MacKinnon, 2016*). All constructs presented here are in the BacMam vector except for the L341 split constructs. For the Eag1TM construct, residues 197–481 (S1-S6) were fused to residues 887–962 (C-terminal tetramer assembly domain). For the Eag1/hErg1 chimera, residues 1–389 (PAS) and 670–1159 (C-linker, CNBHD, and C-terminal assembly domain) of hErg were fused to the N- and C-termini of Eag1 residues 197–481 (S1-S6), respectively. For the L341 split, the N-terminal half (1-341) and C-terminal half (342-963) were each cloned into a pGEM vector for oocyte expression. Mutagenesis and deletions were performed with standard protocols and constructs were confirmed by sequencing. Calmodulin (CaM) was cloned into a BacMam vector as described previously (*Whicher and MacKinnon, 2016*).

## Electrophysiological recordings of Eag1 constructs in CHO cells

All recordings of Eag1 constructs in BacMam vectors were from Chinese hamster ovary (CHO) cells. CHO cells cultured in DMEM-F12 (Gibco) with 10% FBS were transfected with the Eag1 construct using the FuGENE HD transfection reagent (Promega). 48 hr following transfection, the media was replaced with bath solution (10 mM HEPES pH 7.4, 60 mM KCl, 95 mM NaCl, 1 mM CaCl$_2$) and experiments were performed at room temperature using the whole cell patch clamp technique. Polished borosilicate glass pipettes with resistance between 2–4 MΩ were filled with 10 mM HEPES pH 7.4, 165 mM KCl, 5 mM EDTA. To record Eag1 Δ3–13 in the presence of calcium, the 5 mM EDTA was replaced with 1 mM CaCl$_2$ in the pipette solution. Voltage-family recordings were measured by holding the cells at −100 mV, stepping to depolarized voltages up to 100 mV in 20 mV steps, and then stepping back to −80 mV. To determine the V$_{0.5}$ value, normalized tail current vs. voltage was plotted and fit with a Boltzmann function. Cole-Moore effect recordings were measured by holding cells for 500 ms at increasing holding potentials from −190 mV to the voltage of channel activation (either −110 mV, −50 mV or −30 mV depending on the construct) in 20 mV steps followed by a step to 40 mV. To estimate the holding potential that produces half maximal rates of activation (V$_{0.5CM}$), we plotted holding potential vs normalized current at 10 ms following the depolarization step and fit the plot with a Boltzmann function:

$$(I - I_{min})/(I_{max} - I_{min}) = 1/1 + \exp(-ZF/RT(V - V_{0.5cm}))$$

where (I-I$_{min}$)/(I$_{max}$ -I$_{min}$ ) is the normalized current at 10 ms following the depolarization step, V is the hyperpolarization voltage preceding the depolarization step, V$_{0.5cm}$ is the hyperpolarization voltage that produces half maximal rates of activation, F is the Faraday's constant, R is the gas constant, T is the absolute temperature, and Z is the apparent valence of voltage dependence.

To determine reversal potential, we plotted normalized outward current vs depolarization voltage and determined the X intercept. All recordings were measured with pClamp10.5 software (Molecular Devices), an Axopatch 200B amplifier (Molecular Devices), and an Axon digidata 1550 digitizer (Molecular Devices). Recordings were filtered at 1 kHz and sampled at 10 kHz. No leak current was subtracted from the current traces.

## Electrophysiological recordings of Eag1 L341 split constructs in oocytes

The mMessage mMachine T7 transcription kit (Ambion) was used to produce cRNA of the Eag1 split constructs linearized with NdeI. The MEGAclear kit was used to purify cRNAs, which were injected into oocytes. A total of 10 ng of cRNA was injected per oocyte at a ratio of 1:1 N-terminal half:C-

terminal half. Oocytes were stored at 18˚C for 24–48 hr after injection in ND96 (96 mM NaCl, 2 mM KCl, 1.8 mM CaCl$_2$, 1.0 mM MgCl$_2$, 5 mM HEPES pH 7.6 with NaOH, 50 μg/ml gentamycin) and used for recordings. The bath solution was 55 mM NaCl, 60 mM KCl, 1.8 mM CaCl$_2$, and 10 mM HEPES pH 7.2 with NaOH and the pipette solution was 3M KCl. The voltage family protocol was as follows: hold at −20 mV, step to depolarized voltages from −120 to 100 mV in 20 mV steps, and then step back to −80 mV. All recordings were measured at room temperature with pClamp10.5 software (Molecular Devices), Gene Clamp 500 amplifier (Molecular Devices), and an Axon digidata 1440A digitizer (Molecular Devices) in two electrode voltage-clamp configuration. The recorded signal was filtered at 1 kHz and sampled at 10 kHz. No leak or capacitive currents were subtracted from the current traces.

## Expression and purification of Eag1 Δ3–13/CaM

The C-terminal unstructured region (773-886) of the Eag1 Δ3–13 was removed as described previously to improve expression and stability. This mutation does not affect the functional properties of the channel (*Whicher and MacKinnon, 2016*). Baculovirus for Eag1 Δ3–13 and CaM, were produced by transfecting bacmids into SF9 cells in Grace's media supplemented with 10% FBS with the cellfectin II reagent (Invitrogen). Then the baculovirus was amplified in 1L suspension cultures of SF9 cells at 27˚C. 1L cultures of HEK293S GnTI⁻ at $3 \times 10^6$ cells/mL in Freestyle 293 media (Gibco) supplemented with 2% FBS were infected with both Eag1 Δ3–13 and CaM baculovirus at a 4:1 Eag1 Δ3–13: CaM ratio. Following infection, the cells were incubated at 37˚C for 18 hr, induced by adding 10 μM sodium butyrate, incubated at 30˚C for 48 hr, and harvested.

4L of cell pellet was resuspended in lysis buffer (20 mM Tris pH 8, 1 mM CaCl$_2$, 1 μg/ml leupepetin, 1 μg/ml pepstatin, 1 mM benzamidine, 1 μg/ml aprotonin, 0.01 mg/ml DNase, 1 mM PMSF), incubated at RT with stirring for 20 min, and centrifuged for 40 min at 35,000xg. Pellets were resuspended in extraction buffer (50 mM Tris pH 8, 300 mM KCl, 1 mM CaCl$_2$, 8 mM Lauryl Maltose Neopentyl Glycol (LMNG), 2 mM Cholesteryl hemisuccinate (CHS), 1 μg/ml leupepetin, 1 μg/ml pepstatin, 1 mM benzamidine, 1 μg/ml aprotonin, 0.01 mg/ml DNase, 1 mM PMSF), incubated at 4˚C for 2 hr with stirring, and centrifuged for 90 min at 35,000xg. The supernatant was incubated for 2 hr at 4˚C with CNBR-activated sepharose beads (GE healthcare) coupled to a nanobody with high affinity for GFP (GFP-NB) (*Kirchhofer et al., 2010*). The beads were washed with superose 6 buffer (20 mM Tris pH 8, 300 mM KCl, 1 mM CaCl$_2$, 0.05% Digitonin) first with and then without 10 mM MgCl$_2$ and 5 mM adenosine triphosphate (ATP) to remove bound heat shock proteins. The washed beads were incubated overnight at 4˚C with PreScission protease (10:1 w/w ratio) to remove the GFP tag from Eag1 Δ3–13. The protein was eluted with wash buffer, concentrated, and purified on a superose 6 column (GE healthcare) equilibrated with superose 6 buffer. Peak fractions of Eag1 Δ3–13 bound to CaM (Eag1 Δ3–13/CaM) (*Figure 6—figure supplement 1a*) were pooled and concentrated to 5 mg/ml for single particle Cryo-EM structure determination.

## EM sample preparation and imaging of Eag1 Δ3–13/CaM

In a Vitrobot Mark IV (FEI), 3.5 μl of 5 mg/ml Eag1 Δ3–13/CaM was pipetted onto Quantifoil R1.2/1.3 gold holey carbon grids (Quantifoil) with 400 mesh that were glow-discharged for 10 s. The grids were blotted for 4 s at 100% humidity and frozen in liquid nitrogen cooled liquid ethane. Images were collected in a 300keV Titan Krios (FEI) with a Gatan K2 Summit direct electron detector (Gatan) with Serial EM (*Mastronarde, 2005*) in super-resolution counting mode, with a super resolution pixel size of 0.5 Å, and a defocus range of 1.2 to 2.4 μm. Data were collected with a dose of 8 electrons per physical pixel per second (pixel size of 1.0 Å at the specimen) and images were recorded with a 10 s exposure and 200 ms subframes (50 total frames) to give a total dose of 80 electrons per Å$^2$ (1.6 electrons per Å$^2$ per subframe).

## Image processing and map generation

Dose fractionated subframes were binned by 2 (giving a pixel size of 1.0 Å), aligned, and summed using MotionCor2 (*Zheng et al., 2017*) with $5 \times 5$ patches (*Figure 6—figure supplement 1b*). The contrast transfer function was estimated for each summed image using CTFFIND4 (*Rohou and Grigorieff, 2015*). Three projection averages from the previous structure of Eag1 bound to CaM (*Whicher and MacKinnon, 2016*) were used as templates for automated picking in RELION

(*Scheres, 2012*). The automatically selected particles were manually inspected to remove false positives and subjected to 2D classification in RELION specifying 200 classes (*Figure 6—figure supplement 1c*). The lowest populated classes were removed resulting in a data set of 378,000 particles. 3D classification of this data set, with Eag1/CaM as a reference (*Whicher and MacKinnon, 2016*), resulted in five classes with similar numbers of particles and resolution. Therefore, all 378,000 particles were combined for 3D refinement, with C4 symmetry imposed, producing a map at 4.5 Å resolution estimated by gold standard FSC at the 0.143 cutoff criteria (*Scheres and Chen, 2012*). The refined particles were subjected to further rounds of 3D classification without image alignment, which produced 2 subsets of particles: conformation 1 and conformation 2. Conformation 1 of Eag1 Δ3–13/CaM has 43,137 particles and a similar overall structure to Eag1/CaM (*Whicher and MacKinnon, 2016*). Conformation 2 of Eag1 Δ3–13/CaM has 54,530 particles and the intracellular domains are rotated with respect to the transmembrane domains. Bayesian particle polishing and 3D refinement, with C4 symmetry imposed, of the particle subsets in RELION resulted in 3.67 Å for conformation 1 and 4 Å for conformation 2. Gold standard FSC curves were calculated with a mask that excludes the detergent micelle and resolution values were estimated with the FSC = 0.143 cutoff criteria (*Figure 6—figure supplement 1d–f*) (*Scheres and Chen, 2012*). Local resolutions were estimated by ResMap (*Figure 6—figure supplement 2*) (*Kucukelbir et al., 2014*).

## Model building

The models of conformation 1 and conformation 2 were built in Coot (*Emsley et al., 2010*). For both conformations, first the S1-S6 and then the intracellular domains (PAS, C-linker, CNBHD, and CaM) from the structure of Eag1/CaM (*Whicher and MacKinnon, 2016*) (pdb-5K7L) were placed into the density as a rigid body. Following rigid body fitting, the model was manually inspected to fix regions that did not agree with the map or delete regions where there was no density. In conformation 1, we did not observe density for residues 244–246 (S1-S2 linker), 305–322 (S3-S4 linker), 407–411, 697–703, and 721-C-terminus. In conformation 2, we did not observe density for residues 202–213, 243–246 (S1-S2 linker), 274–283 (S2-S3 linker), 305–323 (S3-S4 linker), 407–411, 697–705, and 721-C-terminus. The side chains were modeled as alanine in lower resolution regions. Phenix real space refinement was used to refine the tetramer model of conformation 1 and conformation 2. Final models were validated using MolProbity and by comparing FSCs between the refined model and the EM map (*Figure 6—figure supplement 3*). Figures were generated with Chimera (*Pettersen et al., 2004*), Pymol (The PyMOL Molecular Graphics System, Version 1.8 Schrödinger, LLC.), HOLE (*Smart et al., 1996*), and structure calculations were performed with the SBgrid suite of programs (*Morin et al., 2013*).

## Acknowledgements

We thank Mark Ebrahim and Johanna Sotiris at the Rockefeller University Cryo-EM resource center for help with data collection and Jue Chen and members of the MacKinnon laboratory for helpful discussions. This work was supported in part by National Institutes of Health grant GM43949. JRW is a Damon Runyon Fellow supported by the Damon Runyon Cancer Research Foundation (DRG-2212–15) and RM is an investigator in the Howard Hughes Medical Institute.

The low pass filtered and amplitude modified 3D cryo-EM density maps for Eag1 Δ3–13/CaM conformation 1 (accession code: EMD-20295) and conformation 2 (accession code: EMD-20294) have been deposited in the electron microscopy data bank. Atomic coordinates for Eag1 Δ3–13/CaM conformation 1 (accession code: 6PBY) and conformation 2 (accession code: 6PBX) have been deposited in the protein data bank.

## Additional information

### Funding

| Funder | Grant reference number | Author |
| --- | --- | --- |
| Damon Runyon Cancer Research Foundation | DRG-2212-15 | Jonathan R Whicher |

| National Institutes of Health | GM43949 | Roderick MacKinnon |
| Howard Hughes Medical Institute | | Roderick MacKinnon |

The funders had no role in study design, data collection and interpretation, or the decision to submit the work for publication.

### Author contributions
Jonathan R Whicher, Conceptualization, Formal analysis, Investigation, Methodology, Writing—original draft; Roderick MacKinnon, Conceptualization, Supervision, Funding acquisition, Writing—review and editing

### Author ORCIDs
Jonathan R Whicher (iD) https://orcid.org/0000-0002-3427-2501
Roderick MacKinnon (iD) https://orcid.org/0000-0001-7605-4679

### Decision letter and Author response
Decision letter https://doi.org/10.7554/eLife.49188.025
Author response https://doi.org/10.7554/eLife.49188.026

## Additional files

### Supplementary files
• Transparent reporting form
DOI: https://doi.org/10.7554/eLife.49188.015

### Data availability
The low pass filtered and amplitude modified 3D cryo-EM density maps for Eag1 3-13/CaM conformation 1 (accession code: EMD-20295) and conformation 2 (accession code: EMD-20294) have been deposited in the electron microscopy data bank. Atomic coordinates for Eag1 3-13/CaM conformation 1 (accession code: 6PBY) and conformation 2 (accession code: 6PBX) have been deposited in the protein data bank.

The following datasets were generated:

| Author(s) | Year | Dataset title | Dataset URL | Database and Identifier |
| --- | --- | --- | --- | --- |
| Whicher JR, MacKinnon R | 2019 | Low pass filtered and amplitude modified 3D cryo-EM density maps for Eag1 3-13/CaM conformation 1 | https://www.ebi.ac.uk/pdbe/emdb/EMD-20295 | Electron Microscopy Data Bank, EMD-20295 |
| Whicher JR, MacKinnon R | 2019 | Low pass filtered and amplitude modified 3D cryo-EM density maps for Eag1 3-13/CaM conformation 2 | https://www.ebi.ac.uk/pdbe/emdb/EMD-20294 | Electron Microscopy Data Bank, EMD-20294 |
| Whicher JR, MacKinnon R | 2019 | Atomic coordinates for Eag1 3-13/CaM conformation 1 | http://www.rcsb.org/structure/6PBY | Protein Data Bank, 6PBY |
| Whicher JR, MacKinnon R | 2019 | Atomic coordinates for Eag1 3-13/CaM conformation 2 | http://www.rcsb.org/structure/6PBX | Protein Data Bank, 6PBX |

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
