## [Decision Letter]

Thank you for submitting your article "Regulation of Eag1 gating by its intracellular domains" for consideration by *eLife*. Your article has been reviewed by three peer reviewers, including Gary Yellen as the Reviewing Editor and Reviewer #1, and the evaluation has been overseen by Olga Boudker as the Senior Editor. The following individual involved in review of your submission has also agreed to reveal his identity: Kenton J Swartz (Reviewer #2).

The reviewers have discussed the reviews with one another and the Reviewing Editor has drafted this decision to help you prepare a revised submission.

Summary:

This paper addresses the molecular mechanisms by which the voltage gating of the Eag1 (Kv10.1) channel and other Kv10-12 family members are regulated by their intracellular PAS, C-linker, and CNBHD domains. These channels have a different architecture from the Kv1-9 channels, which couple voltage sensing to gating using an apparently mechanical linkage (the S4-S5 linker). The authors show that the intracellular domains of Eag1 are not required for voltage gating (contrary to their earlier proposal) but that they both substantially modulate the open-closed bias of the channels and can produce the classic "Cole-Moore" effect (a pronounced delay in channel opening after strong hyperpolarization). This extends earlier functional exploration by Pardo and colleagues (2015) and Tomczak et al., 2017, and others, as well as previous structural work by the present authors. The work also addresses the interesting structural and functional analogy between the Cole-Moore effect (delayed opening after strong hyperpolarization) of the Kv10 channels and the slowed deactivation of hErg (Kv11.1) channels.

Essential revisions:

No new experiments are requested, but revisions to the text and figures are important for making the complex arguments here more transparent to the readers, for improving the precision of the explanations, and for providing access to relevant previous studies that bear on the conclusions here.

The paper should include at least one and probably two additional display elements to help explain the proposed model. The Introduction discusses many interesting structural elements in EAG and would greatly benefit from adding a new Figure 1 (or an additional panel) to help the reader see these features. The existing Figure 2 doesn't really suffice for this, and it also doesn't suffice to show all of the residues mutated or deleted. We would suggest a figure that shows everything in detail that is relevant to this study, including residues in the S4-S5 linker or loop. It would also be helpful to include a figure or table that explains the different general gating states essential to thinking about the authors' proposal (something similar to stably shut, pre-open, open, and stably open) and which interactions are thought to occur (for instance PAS-C/Y213/Y639, PAS-N/D342, clockwise or counterclockwise rotation), for both the Eag1 and hErg channels. This would help readers follow the proposal, and it might also help explain more precisely the analogy between the Cole-Moore shift in Eag1 and the slow closure of hErg.

The explanation (in the Discussion, third paragraph) for the slow deactivation of hErg is confusing. The idea is vaguely apparent, but the statement that the slow deactivation in hErg (which arises from open-state stabilization) "arises from the same mechanism as the Cole-Moore effect" (which is a pronounced closed-state stabilization) is not helpful. For the hErg channels, the N-PAS – D342 interaction stabilizes the open state (kinetically if not thermodynamically); for the Eag1 channels, it stabilizes the open state thermodynamically and speeds entry into it. In hErg channels, this produces slow deactivation; in Eag1 channels, the disappearance of this interaction at extreme negative voltages produces the Cole-Moore shift, a kinetic delay in subsequent opening that appears due to the slowness of the re-establishment of this interaction. However, as the authors point out, a simple stabilization in hErg channels does not quite fit, because there is no shift in activation midpoint. Instead, the interaction "prevent<s> pore closing" (but through a kinetic effect). The parallels between the Cole-Moore shift and the slow deactivation are interesting, but the authors should state their case more precisely here.

The authors propose a hypothesis for why the Eag1 delta3-13/CaM structures (the new structures here) remain closed – the Eag1 channels are more biased to closing than the hErg channels [p 10 "A hypothesis for why the pore remains closed in the Eag1 Δ3-13/CaM structure is that, compared to hErg, Eag1 may be more stable in a closed conformation."]. But the fact remains that this exact construct is constitutively open when studied functionally (Figure 6A). So the discrepancy cannot really be explained as a difference between family members – it is clearly a difference between the bilayer and the cryoEM conditions.

In general, please consider alternatives to the proposed model, and explain why you think that your model is better. For instance, the proposed mechanism has the interaction between D342 and the PAS loop as promoting opening by destabilizing closed state interactions between Y213/Y639 and the PAS loop. Instead, or in addition, could the D342-PAS interaction directly stabilize the open state? Or the activated state of the voltage sensor?

Please also address the following questions in the revision, to help clarify the model and the interpretation of the data:

1) If the primary effect of the intracellular domains is to inhibit opening, as suggested in the Abstract ("…provide evidence that VS movement destabilizes these interactions to promote channel opening") and other places in the paper, then why does deletion of intracellular domains, mutation of D342, mutations R7A/R8A, and Δ3-9, destabilize opening (shift the voltage-dependence to the right)?

2) Since there are no estimates of the energetics of the interactions in the structure or the energetic effects of the mutations, the paper should be more cautious about attributing the interactions proposed to the total effects of the intracellular domains.

3) Abstract: "The structure of the calmodulin insensitive mutant suggests that rotation of the intracellular domains promotes channel opening." This sentence is misleading. It should be indicated in the Abstract that the structure presented here is closed, like the original structure of Eag1. It is not clear how a closed structure tells us anything about promoting opening since Ca-CAM doesn't promote opening in Eag1Δ3-13 (does it?). Also, while it seems plausible that the counterclockwise rotation of the intracellular domains is helpful for opening, do we know that it is necessary (it is clearly not sufficient)?

4) Introduction section: "Due to the non-domain swapped transmembrane architecture, the S4-S5 linker is not in the correct position and does not have the required structure to function as a mechanical lever, suggesting an alternative mechanism of voltage-dependent gating in Kvs 10-12." This sentence should be supported by citing previous split experiments.

5) Subsection “Interactions between the voltage sensor and intracellular domains”: "…we first investigated the S4-S5 linker.…by alanine scanning mutagenesis " Wasn't alanine scanning and cysteine scanning mutagenesis previously done on the S4-S5 linker of hErg channels (Gianulis et al., 2013, and Wang et al., 1998 respectively)? This work should be cited.

6) Subsection “Insights into voltage-dependent gating”: "Furthermore, in the open state structure of hErg the PAS loop and the CNBHD Tyr do not interact (Figure 2B)…" What about interaction of the PAS loop and the 213 equivalent residue in the hErg structure?

7) Subsection “Insights into voltage-dependent gating” and elsewhere "…this mechanism provides an explanation for the Cole-Moore effect because at more negative holding potentials the S4 and Asp 342 will be driven further down and away from the PAS loop N-terminus, which will result in slower activation times." Further down from what? Are you suggesting that there is a down state of the VS that is further down than the normal resting state? Wouldn't any movement down prevent interactions of the N-terminal end of the PAS loop with D342?

8) Subsection “Structure of constitutively open Eag1”: "… this opening is transient and thus not observed in the Eag1 Δ 3-13/CaM structure." This is the first mention I found of the previous finding that PAS loop deletions in Eag cause inactivation. This should be mentioned earlier and properly cited.

9) Could the authors discuss why they think the new structures are not inactivated? Could they comment on the position of the S4 helix? Does it appear to be activated as suggested by the earlier CaM-inhibited structure?

10) The IV plots in Figure 5D, 6A and 7D look rather odd because they reverse near -20 mV (the holding voltage) rather than at a voltage near EK^+^. Oocytes normally have internal K^+^ set to 100K^+^, and according to the Materials and methods, these recording were obtained using an external solution with only 2.5 mM K^+^, which puts EK near to -100 mV. The simplest explanation I can think of is that these plots were obtained with much higher external K^+^, perhaps around 50-60 mM, but again the Materials and methods state it is 2.5 mM. Might the authors have plotted the difference between current measured at the holding voltage of -20 mV and each test potential, in some cases then normalized to that difference measured at +100 mV? But this is rather unconventional, and nothing is stated about how this was done. If -20 mV really is the Vrev for these constructs and external K- is 2.5 mM, then either these constructs are no longer K-selective, or the internal concentration of K is way off due to expressing these constructs. Please explain and fix as appropriate. In addition, are the oocytes currents being manipulated in any way? The capacitive currents seem to be largely compensated or subtracted, but nothing is stated in the Materials and methods. Are all the whole-cell CHO currents are not leak subtracted? It would be good to state this specifically. It would also be good to add dashed lines to all traces to indicate the zero-current level. Finally, in the traces shown in Figure 6A, it appears you have lost voltage-clamp in the last few traces. Perhaps replace with another recording or remove those traces?

---

## [Author Response]

Essential revisions:No new experiments are requested, but revisions to the text and figures are important for making the complex arguments here more transparent to the readers, for improving the precision of the explanations, and for providing access to relevant previous studies that bear on the conclusions here.The paper should include at least one and probably two additional display elements to help explain the proposed model. The Introduction discusses many interesting structural elements in EAG and would greatly benefit from adding a new Figure 1 (or an additional panel) to help the reader see these features. The existing Figure 2 doesn't really suffice for this, and it also doesn't suffice to show all of the residues mutated or deleted. We would suggest a figure that shows everything in detail that is relevant to this study, including residues in the S4-S5 linker or loop. It would also be helpful to include a figure or table that explains the different general gating states essential to thinking about the authors' proposal (something similar to stably shut, pre-open, open, and stably open) and which interactions are thought to occur (for instance PAS-C/Y213/Y639, PAS-N/D342, clockwise or counterclockwise rotation), for both the Eag1 and hErg channels. This would help readers follow the proposal, and it might also help explain more precisely the analogy between the Cole-Moore shift in Eag1 and the slow closure of hErg.

A new Figure 1 was added to the manuscript. This figure displays 2 views of the entire Eag1 channel (panel A) and views of the interactions between the PAS loop, VS, S4-S5 linker, C-linker, and CNBHD in the closed state of Eag1 (panel B) and the open state of hErg (panel C). All Eag1 residues that were mutated in this manuscript are shown in panel B.

Figure 1 panels B and C show the interactions that we propose occur in the open (panel C) and closed (panel B) states of Eag1. Instead of creating a new figure or table these figures were referenced when describing the mechanism.

The explanation (in the Discussion, third paragraph) for the slow deactivation of hErg is confusing. The idea is vaguely apparent, but the statement that the slow deactivation in hErg (which arises from open-state stabilization) "arises from the same mechanism as the Cole-Moore effect" (which is a pronounced closed-state stabilization) is not helpful. For the hErg channels, the N-PAS – D342 interaction stabilizes the open state (kinetically if not thermodynamically); for the Eag1 channels, it stabilizes the open state thermodynamically and speeds entry into it. In hErg channels, this produces slow deactivation; in Eag1 channels, the disappearance of this interaction at extreme negative voltages produces the Cole-Moore shift, a kinetic delay in subsequent opening that appears due to the slowness of the re-establishment of this interaction. However, as the authors point out, a simple stabilization in hErg channels does not quite fit, because there is no shift in activation midpoint. Instead, the interaction "prevent<s> pore closing" (but through a kinetic effect). The parallels between the Cole-Moore shift and the slow deactivation are interesting, but the authors should state their case more precisely here.

We agree that the statement that the slow deactivation in hErg "arises from the same mechanism as the Cole-Moore effect" is confusing and it was removed. The data suggest that equivalent residues (D342, the PAS loop N-terminus, and the PAS loop C-terminus) in Eag1 and hErg function in different voltage- dependent gating outcomes. In Eag1 these residues function in the Cole-Moore effect and in hErg these residues function in slow deactivation. In Eag1, mutation of D342 and the PAS loop N-terminus results in a right-shifted V_0.5_ suggesting that these residues are important for channel activation. In hErg, mutation of D540 and the PAS loop N-terminus does not affect V_0.5_ suggesting that these residues are not important for channel activation. Therefore, we propose that in Eag1, D342 and the PAS loop N-terminus interact during channel activation to promote pore opening and thus function in the Cole-Moore effect. Whereas in hErg, D540 and the PAS loop N-terminus interact after channel activation to prevent pore closing and thus function in slow deactivation. In both cases, we believe the interaction between the S4 Asp (342 in Eag1 and 540 in hErg) and the PAS loop N-terminus destabilizes the interaction between the PAS loop C- terminus, the S1 Tyr (213 in Eag1 and 403 in Eag1), and the CNBHD Tyr (639 in Eag1 and 827 in hErg) and thus the closed state of the channel. The paragraph was edited to convey these points.

The authors propose a hypothesis for why the Eag1 delta3-13/CaM structures (the new structures here) remain closed – the Eag1 channels are more biased to closing than the hErg channels [p 10 "A hypothesis for why the pore remains closed in the Eag1 Δ3-13/CaM structure is that, compared to hErg, Eag1 may be more stable in a closed conformation."]. But the fact remains that this exact construct is constitutively open when studied functionally (Figure 6A). So the discrepancy cannot really be explained as a difference between family members – it is clearly a difference between the bilayer and the cryoEM conditions.

The conditions of structure determination, such as detergent solubilization and cryogenic freezing temperatures, can affect a proteins structure. However, the open conformation structure of hErg was determined under similar Cryo-EM conditions with DDM instead of Digitonin as the detergent. In addition, the data presented in this manuscript and previous manuscripts suggest that Eag1 is more stable in a closed conformation than hErg. Therefore, we believe that the closed pore of Eag1 Δ3-13/CaM is likely the result of a combination of Eag1 being more stable in a closed conformation than hErg and the structure determination conditions. This is now reflected in the paper.

“Therefore, we propose that the Eag1 intracellular domains, when viewed from the extracellular side, rotate in a counterclockwise direction to promote the opening of the pore. However, due to the stability of Eag1 in a closed conformation and the conditions under which the Cryo-EM structure was determined we suspect that pore opening is transient and thus not observed in the Eag1 Δ3-13/CaM structure.”

In general, please consider alternatives to the proposed model, and explain why you think that your model is better. For instance, the proposed mechanism has the interaction between D342 and the PAS loop as promoting opening by destabilizing closed state interactions between Y213/Y639 and the PAS loop. Instead, or in addition, could the D342-PAS interaction directly stabilize the open state? Or the activated state of the voltage sensor?

The large right shift in voltage-dependence for the Eag1 TM mutant suggests that the intracellular domains help stabilize the open state of the channel. Therefore, it is possible that the interaction between D342 and the PAS loop N-terminus may promote channel opening by stabilizing the depolarized state of the VS as well as destabilizing the interaction between Y213, Y639, and the PAS loop C- terminus. A few sentences were added to the Results and Discussion sections to account for this possibility.

“How might an interaction between the PAS loop and Asp 342 promote channel opening? One hypothesis is that the interaction between the PAS loop and Asp 342 stabilizes the depolarized state of the voltage sensor. This hypothesis would explain why deletion of the intracellular domains (Eag1TM) and mutation of Asp 342 and Arg 7 and 8 disfavors channel opening as indicated by a right shift in the voltage-dependence of activation.”

Please also address the following questions in the revision, to help clarify the model and the interpretation of the data:1) If the primary effect of the intracellular domains is to inhibit opening, as suggested in the Abstract ("…provide evidence that VS movement destabilizes these interactions to promote channel opening") and other places in the paper, then why does deletion of intracellular domains, mutation of D342, mutations R7A/R8A, and Δ3-9, destabilize opening (shift the voltage-dependence to the right)?

We now explain that upon depolarization we propose that Asp 342 interacts with the PAS loop N-terminus to promote channel opening through the concomitant stabilization of the open state of the VS and destabilization of the interaction between the PAS loop C-terminus, Tyr 213, and Tyr 639 (discussed in the previous comment). Therefore, we would expect deletions of the intracellular domains and mutations to Asp 342 or the PAS loop N-terminus (residues 3-9) to destabilize channel opening.

2) Since there are no estimates of the energetics of the interactions in the structure or the energetic effects of the mutations, the paper should be more cautious about attributing the interactions proposed to the total effects of the intracellular domains.

To the point of the reviewers the description of the mechanism in the discussion was edited as shown:

“Based on the data, we proposed the following general mechanism of modulation of voltage-dependent gating by the intracellular domains. In the depolarized conformation of the VS, Asp 342 interacts with the PAS loop N-terminus to stabilize the depolarized state of the VS as well as destabilize the interaction between the PAS loop C-terminus, Tyr 213, and Tyr 639 to promote channel opening (Figure 1C). In the hyperpolarized conformation of the VS, movement of the S4 disrupts the interaction between Asp 342 and the PAS loop N-terminus to allow for the interaction between the PAS loop C- terminus, Tyr 213, and Tyr 639 to form and promote channel closing (Figure 1B). The structure of Eag1 Δ3-13/CaM in a pre-open conformation suggests that channel opening may occur through a counterclockwise rotation of the intracellular domains and channel closing may occur through a clockwise rotation of the intracellular domains”

3) Abstract: "The structure of the calmodulin insensitive mutant suggests that rotation of the intracellular domains promotes channel opening." This sentence is misleading. It should be indicated in the Abstract that the structure presented here is closed, like the original structure of Eag1. It is not clear how a closed structure tells us anything about promoting opening since Ca-CAM doesn't promote opening in Eag1Δ3-13 (does it?). Also, while it seems plausible that the counterclockwise rotation of the intracellular domains is helpful for opening, do we know that it is necessary (it is clearly not sufficient)?

In the presence of Ca^2+^/CaM Eag1 Δ3-13 is constitutively open. Therefore, the combination of the structure of Eag1 Δ3-13/CaM, which shows a rotation of the intracellular domains compared with the original Eag1 structure, and the open hErg structure strongly suggest that a rotation of the intracellular domains is important for channel opening. However, the sentence in the Abstract may mislead readers into thinking that the structure of Eag1 Δ3-13/CaM is in an open pore conformation. To avoid misleading readers, we have edited the paper to state that the structure of Eag1 Δ3-13/CaM represents a pre-open conformation of the channel. The Abstract was edited as shown:

“The structure of the calmodulin insensitive mutant in a pre-open conformation suggests that channel opening may occur through a rotation of the intracellular domains and calmodulin may prevent this rotation by stabilizing interactions between the VS and intracellular domains.”

4) Introduction section: "Due to the non-domain swapped transmembrane architecture, the S4-S5 linker is not in the correct position and does not have the required structure to function as a mechanical lever, suggesting an alternative mechanism of voltage-dependent gating in Kvs 10-12." This sentence should be supported by citing previous split experiments.

Both Eag1 split channel manuscripts (Lorinczi et al., 2016 and Tomczak et al., 2017) were cited.

5) Subsection “Interactions between the voltage sensor and intracellular domains”: "…we first investigated the S4-S5 linker.…by alanine scanning mutagenesis " Wasn't alanine scanning and cysteine scanning mutagenesis previously done on the S4-S5 linker of hErg channels (Gianulis et al., 2013, and Wang et al., 1998 respectively)? This work should be cited.

These citations were added to the manuscript.

6) Subsection “Insights into voltage-dependent gating”: "Furthermore, in the open state structure of hErg the PAS loop and the CNBHD Tyr do not interact (Figure 2B)…" What about interaction of the PAS loop and the 213 equivalent residue in the hErg structure?

In the open conformation of hErg, Tyr 403 (equivalent to Tyr 213 in Eag1) is near Arg 5 (equivalent to Arg 8 in Eag1) and no longer interacting with the PAS loop C-terminus (Figure 1C). This sentence was added to the manuscript:

“Furthermore, in the open state structure of hErg the PAS loop C-terminus does not interact with either Tyr 403 (equivalent to Tyr 213 in Eag1) or Tyr 827 (equivalent to Tyr 639 in Eag1) (Figure 1C) suggesting that destabilization of this interface might be necessary for channel opening”

7) Subsection “Insights into voltage-dependent gating” and elsewhere "…this mechanism provides an explanation for the Cole-Moore effect because at more negative holding potentials the S4 and Asp 342 will be driven further down and away from the PAS loop N-terminus, which will result in slower activation times." Further down from what? Are you suggesting that there is a down state of the VS that is further down than the normal resting state? Wouldn't any movement down prevent interactions of the N-terminal end of the PAS loop with D342?

The Cole-Moore effect was proposed to be due to the existence of multiple closed states that the VS transitions through in order to reach an active or depolarized conformation. At more negative potentials the VS transitions through more closed states to reach an active conformation, which results in slower activation times. So yes, we (and others before us) think that there are down states of the voltage sensor that are further down than the distribution of states associated with normal resting conditions. In Eag1 specifically, we propose that at more negative holding potentials the VS may have to transition through more closed states in order for the S4 and Asp 342 to interact with the PAS loop N-terminus, which will result in slower activation times. In the absence of a structure, we cannot accurately predict the position of the VS in a hyperpolarized conformation so the phrase “the S4 and Asp 342 will be driven further down and away from the PAS loop N-terminus” was removed from the manuscript. The explanation of the Cole-Moore effect in Eag1 was edited as follows:

“The Cole-Moore effect was proposed to be due to the existence of multiple closed states that the VS must transition through in order to reach an active or depolarized conformation (Cole and Moore, 1960). At more negative potentials the VS must transition through more closed states to reach an active conformation, which results in slower activation times. In Eag1, we propose that at more negative holding potentials the VS might have to transition through more closed states in order for the S4 and Asp 342 to interact with the PAS loop N-terminus, which will result in slower activation times.”

8) Subsection “Structure of constitutively open Eag1”: "… this opening is transient and thus not observed in the Eag1 Δ 3-13/CaM structure." This is the first mention I found of the previous finding that PAS loop deletions in Eag cause inactivation. This should be mentioned earlier and properly cited.

This is cited in subsection “Implications for voltage-dependent gating” when discussing PAS loop deletions.

9) Could the authors discuss why they think the new structures are not inactivated? Could they comment on the position of the S4 helix? Does it appear to be activated as suggested by the earlier CaM-inhibited structure?

The Eag1 Δ3-13 does not inactivate at 0 mV when bound to Ca^2+^/CaM (Figure 6A) so we would not expect the structure of Eag1 Δ3-13/CaM to be in an inactivated conformation.

The S4 helix is in depolarized conformation similar to the original structure of Eag1 bound to CaM. This was included in the manuscript:

“In both conformations, the S4 helices adopt a depolarized conformation and intracellular domains are rotated in a counterclockwise direction when viewed from the extracellular side of the membrane but in conformation 2 the extent of the rotation is larger (2.4° degrees for conformation 1 and 8.6° degrees for conformation 2) (Figure 6B,C).”

10) The IV plots in Figure 5D, 6A and 7D look rather odd because they reverse near -20 mV (the holding voltage) rather than at a voltage near EK^+^. Oocytes normally have internal K^+^ set to 100K^+^, and according to the Materials and methods, these recording were obtained using an external solution with only 2.5 mM K^+^, which puts EK near to -100 mV. The simplest explanation I can think of is that these plots were obtained with much higher external K^+^, perhaps around 50-60 mM, but again the Materials and methods state it is 2.5 mM. Might the authors have plotted the difference between current measured at the holding voltage of -20 mV and each test potential, in some cases then normalized to that difference measured at +100 mV? But this is rather unconventional, and nothing is stated about how this was done. If -20 mV really is the Vrev for these constructs and external K- is 2.5 mM, then either these constructs are no longer K-selective, or the internal concentration of K is way off due to expressing these constructs. Please explain and fix as appropriate.

A higher external K^+^ concentration of 60mM was used for these experiments. This is now updated in the Materials and methods section.

In addition, are the oocytes currents being manipulated in any way? The capacitive currents seem to be largely compensated or subtracted, but nothing is stated in the Materials and methods.

No leak or capacitive currents were subtracted from the oocyte current traces. This was added to the Materials and methods.

Are all the whole-cell CHO currents are not leak subtracted? It would be good to state this specifically.

The whole-cell CHO current traces are not leak subtracted. This was added to the Materials and methods.

It would also be good to add dashed lines to all traces to indicate the zero-current level.

Dashed lines to indicate zero current level were added to all traces.

Finally, in the traces shown in Figure 6A, it appears you have lost voltage-clamp in the last few traces. Perhaps replace with another recording or remove those traces?

The last few traces were removed from this figure.